# Changes in Key Aroma Compounds and Esterase Activity of *Monascus*-Fermented Cheese across a 30-Day Ripening Period

**DOI:** 10.3390/foods11244026

**Published:** 2022-12-13

**Authors:** Hong Zeng, Yadong Wang, Haoying Han, Yanping Cao, Bei Wang

**Affiliations:** School of Food and Health, Beijing Technology and Business University, Beijing 100048, China

**Keywords:** esterase, gas chromatography–olfactometry–mass spectrometry (GC-O-MS), sensory analysis

## Abstract

*Monascus*-fermented cheese (MC) is a new type of mold-ripened cheese that combines a traditional Chinese fermentation fungus, *Monascus purpureus* M1, with Western cheese fermentation techniques. In this study, the compositions of the volatile aroma compounds in MC were analyzed during a 30-day ripening period using SPME-Arrow and GC-O-MS. The activity of esterase in MC, which is a key enzyme catalyzing esterification reaction, was determined and compared with the control group (CC). Next, sensory analysis was conducted via quantitative descriptive analysis followed by Pearson correlation analysis between esterase activity and the key flavor compounds. A total of 76 compounds were detected. Thirty-three of these compounds could be smelled at the sniffing port and were identified as the key aroma compounds. The esterase activity in MC was found to be 1.24~1.33 times that of the CC. Moreover, the key odor features of ripened MC were alcohol and fruity flavors, considerably deviating from the sour and cheesy features found for the ripened CC. Furthermore, correlation analysis showed that esterase activity was strongly correlated (|r|> 0.75, *p* < 0.05) with various acids such as pentanoic and nonanoic acids and several aromatic esters, namely, octanoic acid ethyl ester and decanoic acid ethyl ester, revealing the key role that esterases play in developing the typical aroma of ripened MC.

## 1. Introduction

*Monascus* sp. has been acknowledged as an industrially important mold due to its ability to produce various valuable metabolites, including red pigments, *γ*-aminobutyric acid, and monacolin K [1]. Research on *Monascus* has been centered on its potential therapeutic uses for curing several chronic diseases, such as hyperlipemia, hypertension, and dengue virus infection [2]. In addition to its potential clinical benefit, it is worth noting that *Monascus* sp. has also been used as a starter culture in traditional Chinese fermented foods (e.g., *Monascus* rice wine, *Monascus* vinegar, and red fermented bean curd) for over 1000 years [3]. Ren et al. successfully isolated and screened 35 fermentation starters among the *Monascus* strain FJMR24 with high amylase activity and high glucoamylase activity, resulting in outstanding fermentation adaptability in the process of rice wine brewing [4]. Studies on *Monascus* vinegar, on the other hand, mostly focus on its lipid-lowering effect [5]. Jia Song et al. [6] reported the lipid-lowering mechanism of *Monascus* vinegar and its correlation with gut microbiota, degree of lipid metabolism, and inflammation. Their study suggests that *Monascus* vinegar is beneficial for preventing cardiovascular diseases.

Although *Monascus* sp. has been well studied in rice wine and vinegar products, studies on its application in cheese-making are relatively scarce. To our knowledge, the research on *Monascus*-fermented cheese (MC), which is a new type of mold-ripened cheese, mainly focuses on the optimization of the production process and the physicochemical properties of the cheese [7,8]. For instance, *M. purpureus* BD-M-4 has been used as the secondary starter for producing a novel semi-hard cheese, and has been reported to possess proteolytic activity for αs_1_-casein and *β*-casein [9]. Xia et al. produced an internal mold-ripened cheese using *Monascus* fumeus x08 and characterized its proteolysis, lipolysis, textural, and sensory properties. The flavor and taste intensity of MC are significantly lower than blue cheeses currently on the market; however, the color intensity and overall receptivity are significantly higher [10]. From this point of view, *Monascus* sp. could carry high market value for cheese manufacturers.

Previous studies have shown that enzymatic reactions and the resulting chemical changes that take place during the ripening process have important impacts on the final aroma and sensory properties of cheeses [11,12]. *Monascus* possesses various esterases, proteases, and lipases that can be excreted to the extracellular space to accelerate the hydrolysis of protein and fat in dairy products. The role of protease and lipase in accelerating cheese ripening and promoting the formation of aroma compounds has been extensively studied [13,14]. The use of esterases in the cheese production process, together with a milk-clotting preparation technique containing a certain ratio of proteolytic enzymes (chymosin/pepsin), has been proven to positively affect the sensory properties and shorten the maturation time of cheeses [15]. Nevertheless, how esterase converts cheese components into aroma compounds through specific metabolic reactions remains unclear.

In this study, *Monascus purpureus* M1 spore solution was used as an adjunct starter to produce mold-ripened cheese. The volatile aroma compounds, esterase activity and sensory properties of *Monascus*-fermented cheeses ripened for 30 days were analyzed and compared with a control group, i.e., cheeses fermented without *Monascus*. The key objective was to investigate how *Monascus*, and in particular esterases, affect the formation of volatile aroma compounds and change sensory properties during the ripening period. This study is expected to provide engineering ideas for the regulation of the final aroma of *Monascus*-fermented cheese products.

## 2. Materials and Methods

### 2.1. Cheese Manufacture

The raw milk used in the experiment was provided by a local dairy farm (Junlebao Co., Ltd., Shijiazhuang, Hebei, China). The lactose, protein, fat, and pH were 4.65%, 3.35%, 3.81% and 6.56 in the raw milk, respectively. The preparation scheme of CC and MC was conducted according to Ai et al. [10], with some modifications (Figure 1). After standardization (fat content, 3.4 g/L), the milk was pasteurized at 65 °C for 30 min. Before renneting, the milk was inoculated with a commercial freeze-dried starter culture (CHR R-704) and *Monascus purpureus* M1 spore liquid (5.0 × 10^7^ spores per milliliter) at 31 °C and acidified for 60 min, while the control cheeses (CC) were produced with a conventional commercial starter. Then, commercial rennet was added (0.2 g/L milk) at 31 °C for 45 min. The curd was cut into approximately 3 × 3 × 3 cm cubes. The curd was healed for 2 min at 31 °C before being stirred for 10 min with a glass rod. Subsequently, the curd was gradually cooked to 38 °C over 30 min, and the water bath was slowly increased to 39 °C. The curd was stirred with a glass rod at 38 °C until the pH reached 6.2. After draining off the whey, NaCl (2.00 g/100 g curd) was added to the curd. Then, the curd was placed into square molds and pierced with rods to create tunnels in the curd. Finally, the cheeses were transferred to an incubator (KB450, Binder, Neckarsulm, Germany) at 26 °C and 70% relative humidity for 5 days, and then were kept at 8 °C for 25 days. Samples from the cheeses were taken at seven different ripening periods (0 day, 5 day, 10 day, 20 day, and 30 day) for further analysis. Samples were taken in triplicate.

### 2.2. Gas Chromatography–Olfactometry–Mass Spectrometry (GC-O-MS) Analysis

The GC-MS method was conducted according to Wang et al. [16], with some modifications. Six grams of ground cheese sample was transferred into a 40/mL glass vial with a silicon septum (Uno Instrument Co., Ltd., Zhengzhou, China). The prepared sample was placed into a thermostatic water bath and equilibrated for 30 min at 60 °C. Then, a 20 mm × 120 μm divinylbenzene/carbon wide range/polydimethylsiloxane fiber (DVB/CWR/PDMS, Supelco, Bellefonte, PA, USA) was exposed to the sample headspace for 30 min at 60 °C. After extraction, the solid-phase microextraction (SPME)-Arrow fiber was transferred to the injector port and desorbed for 5 min at 230 °C. The injector temperature was 250 °C.

The GC-O-MS analysis was performed on an Agilent 7890B GC equipped with an Agilent 5977A mass selective detector and an olfactory detection port (Gerstel ODP-3, Mulheim a der Ruhr, Germany). The SPME-Arrow was inserted into the injector port using the splitless mode. Samples were analyzed on a DB-Wax column (60 m × 0.25 mm i.d. × 0.25 μm film, Agilent Technologies, Santa Clara, CA, USA), and the injector port was held at 230 °C. The column carrier gas was helium at a constant flow rate of 1.2 mL/min. The oven temperature was held at 40 °C for 5 min and increased to 230 °C at a rate of 3 °C/min. The mass spectrometer was operated in electron impact mode with 70 eV ionization energy, and the source temperature was set at 230 °C. The full-scan acquisition was used in the 30 to 350 *m*/*z* range. High-purity helium (99.99%) was applied as the carrier gas for GC-O-MS analysis. Each analysis was performed in triplicate.

To avoid the potential loss of odor-active compounds, gas chromatography–olfactometry (GC-O) analyses were carried out by 4 well-trained panelists. During the GC-O analysis [17], the panelists recorded the aroma descriptor and the time when the aroma could be smelled. If 2 or more panelists detected the aroma, an odor-active location was identified. Before the experiments, these panelists underwent training that involved sniffing 60 reference compounds (prepared as 3- to 5-fold odor thresholds in water or odorless sunflower oil [18]) determined by pre-experiments of MC.

n-Alkanes (C_7_~C_30_) were analyzed under the same conditions to calculate the retention indices (RIs) for the volatile compounds. The compounds were identified by comparing the RIs with those described in the literature by comparing their mass spectra with those contained in the NIST 14 libraries. The qualitative analysis was further performed using odor description (O) and verified by a standard compound (S).

In this work, the relative odor activity value (ROAV) was used to describe the intensity of the flavor compounds in the samples [19]. ROAV is defined as the ratio of the relative concentration of a single compound to the odor threshold of that compound [20]. The thresholds were drawn from the literature [18].The larger the ROAV, the greater the contribution of the compound to the overall flavor of the complex flavor mixture. When the ROAV is >1, a compound is considered to contribute to overall aroma.

### 2.3. Esterase Activity

The method was conducted according to Mukdsi et al. [21], with some modifications. The cheese (30 g) was added to 60 mL of 50 mM acetic acid buffer (pH 5.0). It was then dispersed for 1 min with a high-speed disperser, and 40 mL of ethanol was added to fully mix it. Then, the supernatant was centrifuged as the crude enzyme solution for esterase activity determination.

Subsequently, 0.2 mL of enzyme solution, 10 mL of cyclohexane, 3.55 mL of ethanol and 6.25 mL of hexanoic acid were mixed and put into a 100 mL closed conical flask for esterification at 35 °C. After 24 h of reaction, 0.5 mL of supernatant was taken and put into a 50 mL conical flask. Then, 5 mL of water was added. Phenolphthalein was used as the indicator. Titrate with 0.1 mol/L NaOH was added to the end point in order to determine the consumption of hexanoic acid in the esterification process. The amount of enzyme required to consume 1 μmol hexanoic acid per hour at 35 °C is 1 enzyme activity unit (U).

### 2.4. Sensory Evaluation

The sensory evaluations were performed by 10 panelists (2 men and 8 women, with an average age of 23 years), who had previously received professional training in sensory evaluation. The sensory attributes of CC and MC were assessed using quantitative descriptive analysis (QDA). The flavor descriptors (yogurt, creamy, fruity, cheesy, sour, and alcohol) were chosen in the preliminary tests by the panelists [22]. The 6 descriptors were defined as follows, and relevant material objects or chemical standards were used as references [23,24,25]: 10 mg/kg solution in distilled water diacetyl for “yogurt”; cream with 35.5% fat content for “creamy”; 20 mg/kg ethyl hexanoate solution in distilled water for “fruity”; fresh butter for “cheesy”; 0.08% citric acid solution in distilled water for “sour”; 10 mg/kg n-butanol and 10 mg/kg ethyl caproate dissolved in distilled water for “alcohol”. The 6 g cheese samples or recombinants were kept in a flavor proof container (100 mL) until each panelist could give their evaluation. The flavor properties were rated from 0 (none) to 3 (very strong). The tests were performed in triplicate.

### 2.5. Statistical Analysis

The data of the present study were subjected to analysis of variance using IBM SPSS 23.0 (SPSS Inc., Chicago, IL, USA). The aroma compound compositions, sensory scores, and the esterase activity of *Monascus*-fermented cheeses during ripening using one-way ANOVA followed by Duncan’s-test. The statistical significance of the difference was taken as *p* ≤ 0.05. Additionally, Pearson correlation analysis between aroma compounds and esterase activity was performed using the OriginPro 2021 64-bit (OriginPro Lab Corp., Northampton, MA, USA).

## 3. Results and Discussion

### 3.1. Analysis of Volatile Aroma Compounds

SPME-Arrow combined with the GC-O-MS method was used to extract and detect the volatile aroma compounds of CC and MC. As shown in Figure 2, 76 volatile aroma compounds were detected in MC, and 33 of these 76 aroma compounds were detected in CC. Among the 76 volatile aroma compounds, 33 compounds could be smelled at the sniffing port (Appendix A). These 33 volatile aroma compounds were identified as the key aroma compounds of CC and MC, including 10 acids, 7 ketones, 2 alcohols, 12 esters and 2 aromatic heterocyclic compounds. To further analyze the contribution of different types of key volatile compounds to the overall aroma, the relative odor activity values (ROAV) of the 33 key aroma compounds were determined (Table 1), as discussed below.

#### 3.1.1. Acids

Short-chain and mid-chain acids are commonly considered key contributors to aroma development in cheeses [16,26]. They are not only the main aroma substances in cheeses, but also the precursors of other aroma substances such as esters, aldehydes, and methyl ketones [27]. In this work, the acids detected by GC-O-MS were acetic acid, butanoic acid, pentanoic acid, hexanoic acid, heptanoic acid, octanoic acid, nonanoic acid, decanoic acid, undecanoic acid, and dodecanoic acid. Of these, decanoic acid reached the highest ROAV on day 10 (225.37). These volatile acids generally result in cheeses with unpleasant, pungent and milky flavors. As the ripening time increased, the ROAV of all detected acids increased until day 20, when this value decreased for both CC and MC. Before day 5, the ROAVs of the acids in MC and CC were relatively comparable. On day 10, the contents of heptanoic acid and decanoic acid in MC increased significantly, and remained two orders of magnitude larger than CC. This variation was consistent with the results of You et al. [9], where MC was reported to contain more short- and intermediate-chain acids than CC. The increase in acids could have resulted from the rapid growth in starter and adjunct microbes cultured at relatively high temperatures at the early stage of ripening. During fast growth, the activity levels of proteases and lipases are high, which can accelerate the hydrolysis of proteins and fat in cheeses [10], and thus lead to the accumulation of short-chain fatty acids. When the contents of acids with good ester synthesis ability are high enough, esterification reactions will be activated, and can contribute to a decrease in acids in the later stage of ripening [28].

#### 3.1.2. Ketones

Most of the ketones identified in MC were methyl ketones, which are generally synthesized from fatty acids. In microorganisms, fatty acids can be oxidized by the enzyme to generate *β*-keto acid. The decarboxylation of *β*-keto acid, i.e., the loss of one carbon atom, produces methyl ketone [29]. Methyl ketones play a key role in the aroma development of mold-ripened cheeses due to their typical flavor and low flavor threshold [29], which is the most abundant volatile compound in mold-ripened cheeses’ neutral compound [30]. The ROAVs shown in Table 1 can be used as indirect indicators of compound contents. The contents of hexanoic acid, heptanoic acid, octanoic acid, nonanoic acid, decanoic acid, undecanoic acid, and dodecanoic acid in MC are higher than in CC during the first 10 days of ripening; meanwhile, the contents of 2-pentanone, 2-heptanone, 3-hydroxy-2-butanone, 2-octanone, 2-nonanone, 2-decanone, and 2-undecanone also increased. The ketone with the highest ROAV is 2-undecanone on day 10 (785.93), followed by 2-nonanone on day 10 (631.20), both of which have a typical fruit flavor.

#### 3.1.3. Esters

Esters can provide floral and fruity flavors and reduce the spicy and bitter flavors of fatty acids and amines [31]. Especially, ethyl esters containing short- to medium-chain fatty acids often impart a pleasant smell. It has been reported that the main routes of ethyl ester synthesis in cheeses are esterification and alcoholysis reactions catalyzed by esterases and alcohol acyltransferases [32]. In addition, Malcata et al. found that with the participation of some lactic acid bacteria, yeasts, or molds, ethyl esters can be synthesized from other pathways, such as acidolysis and transesterification [33]. As shown in Table 1, the ROAVs of the ester compounds in both MC and CC increased with ripening time. The ROAV gradually stabilized after 20 days and then decreased. The hexanoic acid ethyl ester on day 20 (645.39) has the highest ROAV compared with the other esters. MC has a higher ester content than CC, which could result from the presence of various esterases in *Monascus* that produce esters by decomposing fatty acids [34,35]. Furthermore, MC also contains more alcohol compounds, which are precursors of ester biosynthesis. These factors together can explain the higher ester content in MC. The increase in ester content in MC coincides with the fact that the addition of adjunct culture would be beneficial to enhance ester formation in cheese [36].

#### 3.1.4. Alcohol and Aromatic Heterocyclic Compounds

In fermented foods, alcohols are mainly derived from four metabolic pathways: lactose metabolism, methyl ketone reduction, amino acid metabolism, and linoleic acid and linolenic acid degradation [37]. The alcohols that can be smelled in MC and CC are 2-nonanol and benzene ethanol (Table 1). 2-Nonanol has creamy and fruity flavors. The ROAV of 2-nonanol in MC reached the maximum value (3.81) on day 10 and then gradually decreased. Benzene ethanol has a typical floral and honey aroma. The ROAV of benzene ethanol in MC reached the maximum value (3.45) on day 10. The aromatic heterocyclic compounds that can be smelled are benzaldehyde and styrene, both of which are related to the conversion of tryptophan and phenylalanine. Benzaldehyde has a nutty flavor, which plays an important role in the overall good aroma of cheese [38]. Benzaldehyde did not change significantly during ripening. Styrene has a plastic flavor, and thus a high content of styrene can result in undesirable aromas.

### 3.2. The Analysis of Esterase Activity

The main role of esterases in fermented dairy products is to hydrolyze triglycerides from milk fat and release fatty acids [39]. Fatty acids such as butanoic acid, hexanoic acid, and octanoic acid are responsible for the typical spicy flavors of cheeses. Esterases can also catalyze ester synthesis under certain environmental conditions [39]. In this work, to evaluate the role of esterase in cheese matrices, specific esterase activities were determined for cheeses ripened for 0, 5, 10, 20, and 30 days. It was observed that, as time extended, the esterase activity of MC and CC first increased, and then decreased. The highest activity was obtained on day 20 (Figure 3a). Notably, on day 5, the esterase activities of MC became higher than that of CC, and increased 1.24~1.33 times as ripening proceeded. A significant increase in esterase activity during ripening has also been found in studies of other varieties of cheeses, for instance, goat’s cheese [39] and camembert cheese [36].

The ester content of cheeses depends on the balance between ester synthesis reactions and fat hydrolysis reactions. Linking back to the results shown in Section 3.1.1 and Section 3.1.3, the content of acid, which is one of the major precursors of esters, increased rapidly during the first 10 days of ripening, and remained relatively stable from day 10 to day 20, before finally decreasing from day 20 to day 30. Consistently, the content of ester compounds reached the maximum on the 20th day of ripening. These results suggest that, in the early stage of ripening, the hydrolysis of fat prevails. Over the course of the ripening, acids gradually accumulate to a point that activates ester synthesis, which consumes acids and leads to the subsequent downward trend.

In addition, a consistent variation between total acids and esters and esterase activity was also recognized. It can be seen from Figure 3a,b that, from day 0 to day 10, the content of total acids and total esters of MC increased as the esterase activity increased. Moreover, the increasing rate of acids and esters (i.e., the slope of the curve) of MC in this period was high, which coincided with the high level of ester activity. From day 10 to day 20, the esterase activity and the content of total esters of MC continued to increase, but at a lower rate. It is worth noting that, in this period, the content of total acids began to decline, which could have resulted from the activation of esterification reactions (as discussed above). Finally, from day 20 to day 30, both the total acids and esters decreased as the esterase activity decreased. These observations additionally prove that the release of fatty acids and esters in cheeses is strongly linked to esterases during ripening [40].

### 3.3. The Analysis of Sensory Properties

The sensory properties of CC and MC were assessed using quantitative descriptive analysis (QDA) at five different ripening times (0 d, 5 d, 10 d, 20 d, and 30 d). During the 0–10-day period, the key odor features (defined by the top two sensory properties) of CC and MC were identical, i.e., both having yogurt and creamy flavors. After day 10, the cheese flavor became more complex and the key odor features of CC and MC started to deviate from each other (Figure 4).

Firstly, for CC, as the ripening time proceeded after day 10, the content of free fatty acids, especially those with short and medium carbon chains (e.g., butanoic acid, nonanoic acid, and octanoic acid, with reference numbers 2, 6, and 7, as shown in Table 1 and Section 3.1.1), continued to increase, which could have contributed to the strong sour and cheesy flavors of CC during the 10–30-day period. It has been reported that when the sour odor intensity is high, it can mask creamy and yogurt flavors [41]. In this work, we indeed observed a decreased trend in creamy and yogurt flavors (which mainly come from ketones and some esters [42]) with increasing sour and cheesy flavors (Figure 4a,c).

In contrast to the sour and cheesy flavors of CC, the key odor features of MC from day 10 to day 30 were alcohol and fruity flavors (Figure 4b,d). In cheeses, alcohol and fruity flavors are mainly derived from ester compounds [31]. Correspondingly, the content of esters in MC, especially ethyl esters that have alcohol and fruity flavors, increased and reached the highest level on day 20 (e.g., hexanoic acid, ethyl ester, heptanoic acid, ethyl ester, octanoic acid, and ethyl ester, with reference numbers 20, 21, and 23 as shown in Table 1 and Section 3.1.3). After 20 days, as the content of ester compounds in MC began to decrease, the alcohol and fruity flavors in MC also weakened synchronously.

### 3.4. Correlation Analysis between Volatile Aroma Compounds and Esterase Activity

To further investigate the role that esterases played in shaping the final aroma of MC, Pearson correlation coefficients were calculated between the 33 key aroma compounds and esterase activity (Figure 5). Twenty aroma compounds were found to be significantly correlated with esterase activity (|r| > 0.6, *p* < 0.05), including nine acids, three methyl ketones, and eight esters. These compounds may contribute to the formation of the special aroma of *Monascus*-fermented cheeses.

The correlation analysis showed that the esterase activity had an important contribution to a variety of fatty acids. For instance, the esterase activity showed a strong positive correlation (r > 0.8) with butanoic acid, pentanoic acid, hexanoic acid, heptanoic acid, octanoic acid, and nonanoic acid. This coincided with the fact that esterases can be excreted as extracellular enzymes by starter and adjunct microorganisms, which promotes the release of short-chain fatty acids [43].

Furthermore, ethyl ester compounds such as hexanoic acid ethyl ester, octanoic acid ethyl ester, decanoic acid ethyl ester and undecanoic acid ethyl ester and methyl ketones such as 2-nonanone, 2-decanone, and 2-undecanone are strong correlated with esterase activity. Fatty acids are precursors of esters and methyl ketones, which explains the strong correlation that the esterase possesses with esters and methyl ketones. Esterase can utilize fatty acids that derive from fat hydrolysis, protein hydrolysis and lactose fermentation as substrates for ester synthesis [44]. Fatty acids can be converted to methy ketones via *β*-oxidation [29]. The strong correlation between esterase and acid compounds also indirectly suggests that esters and methyl ketones are affected by esterase activity. It can be preliminarily concluded that the presence and high abundance of esterase-harbored *Monascus* in the MC is one of the main reasons for the observed high contents of esters and methyl ketones. Since most esters and methyl ketones provide positive aromas, they are also related to the better sensory properties of MC. In summary, esterase plays a significant role in elevating the aromas and sensory properties of cheeses.

In the future, we hope to elucidate the metabolic pathways behind key aromatic compounds and regulate esterase activity to obtain a better aroma profile of MC using metabolomic and proteomic analysis.

## 4. Conclusions

This work investigated the volatile aroma compounds, the sensory properties and their associations with the esterase activity of *Monascus*-fermented cheeses (MC) across a 30-day ripening period. The optimal ripening time of MC was found to be 20 days according to the overall sensory score and the evolution of 33 key aroma compounds. The esterase activity reached its peak on day 20, correspondingly. Furthermore, Pearson correlation analysis showed that esterase activity was not only strongly correlated with pentanoic and nonanoic acids, octanoic acid ethyl ester, and decanoic acid ethyl ester, but also closely tied with 2-nonanone and 2-decanone. This suggest that esterases released from *Monascus* played a vital role in the utilization of acids as well as the synthesis of ethyl esters and methyl ketones. The current work elucidates that further regulations of esterases can enhance the intensity of cheese flavor, which could be achieved by adjusting the esterase activity to a level that boosts ester production and reduces the content of pungent acids without violating ketone synthesis, to obtain a more balanced aroma of mold-ripened cheeses.

## 5. Patents

New *Monascus purpureus* with high esterase production, useful for producing milk cheese. (CN111423986).

## Figures and Tables

**Figure 1 foods-11-04026-f001:**
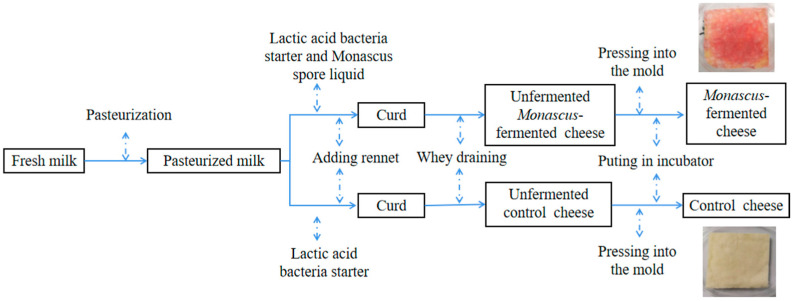
The process of making the *Monascus*-fermented cheese and the control cheese.

**Figure 2 foods-11-04026-f002:**
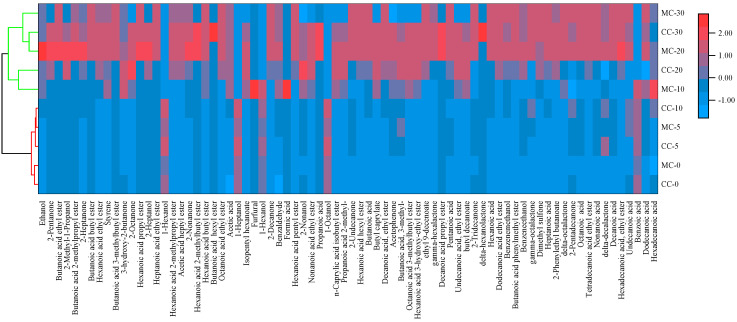
Heat map of the relative contents of volatile aroma compounds in CC and MC during the 30-day ripening period. Data presented in the heat map are in the log10 scale. CC and MC represent cheese samples collected from control cheeses and *Monascus*-fermented cheeses, respectively. CC-0: cheese samples of the control group ripened for 0 days.

**Figure 3 foods-11-04026-f003:**
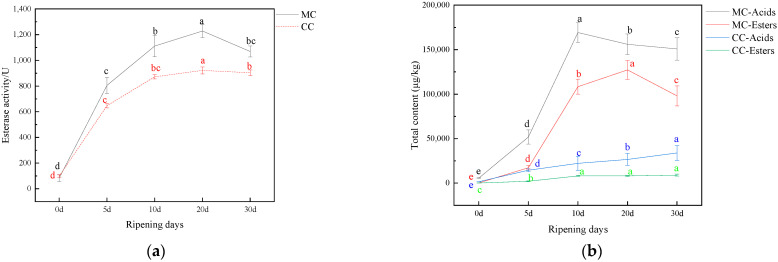
(**a**) The activity of esterases of CC and MC during the 30-day ripening period; (**b**) the content of total acids and total esters of CC and MC during the 30-day ripening period. CC and MC represent cheese samples collected from control cheeses and *Monascus*-fermented cheeses, respectively. (Different letters in the same line correspond to statistically significant differences (*p* < 0.05)).

**Figure 4 foods-11-04026-f004:**
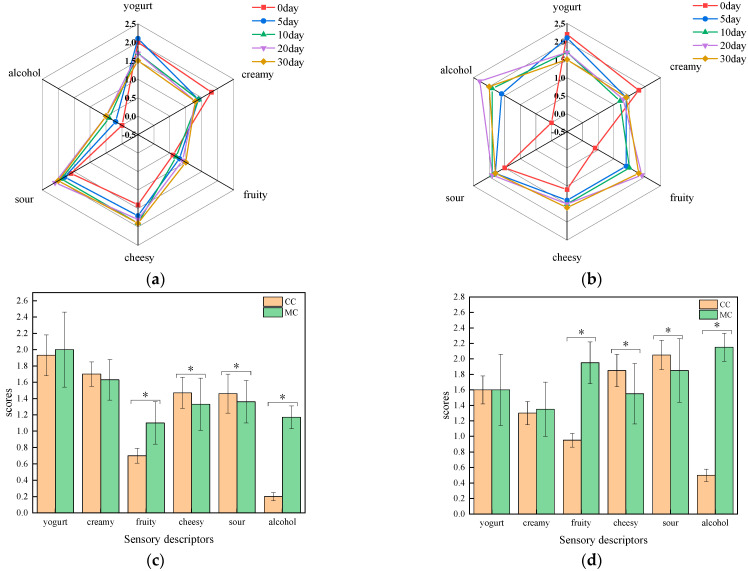
The sensory properties of CC and MC during the 30-day ripening period. (**a**) Radar graph displaying sensory evaluation results of CC; (**b**) radar graph displaying sensory evaluation results of MC; (**c**) the average scores of sensory descriptors of CC and MC during the 0–10-day period; (**d**) the average scores of sensory descriptors of CC and MC during the 10–30-day period. CC and MC represent cheese samples collected from control cheeses and *Monascus*-fermented cheeses, respectively (* corresponds to statistically significant differences (*p* < 0.05)).

**Figure 5 foods-11-04026-f005:**
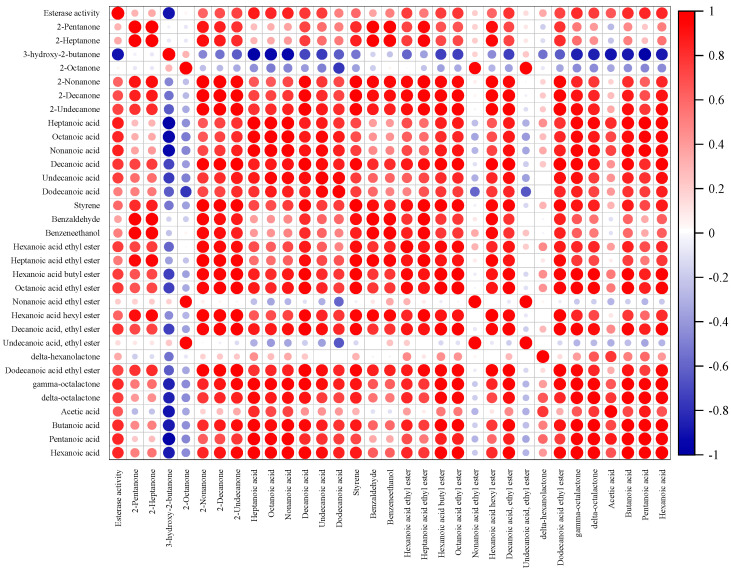
Correlation matrix between the 33 key aroma compounds and the esterase activity of MC. All the Pearson coefficients shown in the matrix are significant (*p* < 0.05).

**Table 1 foods-11-04026-t001:** The ROAV of 33 key aroma compounds identified in CC and MC during the 30-day ripening period. (Different letters in the same row correspond to statistically significant differences (*p* < 0.05)).

No.	Compound	RI			Threshold [18] (μg/kg)	ROAV										Flavor Description
		Calc ^1^	Ref ^2^	ID ^3^												
	**Acids**					**CC-0**	**CC-5**	**CC-10**	**CC-20**	**CC-30**	**MC-0**	**MC-5**	**MC-10**	**MC-20**	**MC-30**	
1	Acetic acid	1450	1449	MS,RI,O,S	700	1.20 ^b^	1.21 ^b^	1.23 ^b^	1.26 ^b^	1.59 ^a^	1.22 ^b^	1.23 ^b^	1.28 ^b^	1.76 ^a^	1.60 ^a^	Pungent, sour, vinegar.
2	Butanoic acid	1621	1637	MS,RI,O,S	410	5.52 ^e^	6.58 ^d,e^	10.49 ^d^	9.43 ^d^	15.84 ^c^	4.26 ^e^	9.37 ^d^	76.95 ^a^	82.32 ^a^	62.60 ^b^	Sharp, rancid, butter.
3	Pentanoic acid	1738	1762	MS,RI,O,S	1700	0 ^c^	0 ^c^	0 ^c^	0.01 ^c^	1.02 ^c^	0.01 ^c^	0.02 ^c^	1.43 ^b^	2.16 ^a^	1.85 ^a^	Rancid
4	Hexanoic acid	1852	1861	MS,RI,O,S	890	0.92 ^d^	1.00 ^d^	1.12 ^d^	1.12 ^d^	1.4 ^c^	1.28 ^cd^	2.46 ^b^	6.74 ^a^	6.27 ^a^	5.73 ^a^	sour, fatty.
5	Heptanoic acid	1954	1950	MS,RI,O,S	100	0.16 ^c^	0.11 ^c^	0.22 ^c^	0.22 ^c^	0.38 ^c^	0.14 ^c^	3.03 ^b^	22.18 ^a^	27.34 ^a^	28.24 ^a^	Rancid, sour.
6	Nonanoic acid	2065	2050	MS,RI,O,S	4600	0 ^c^	0 ^c^	0.12 ^c^	1.01 ^b^	0.09 ^c^	0 ^c^	0.01 ^c^	1.25 ^a^	1.18 ^b^	0.21 ^c^	Waxy, fatty.
7	Octanoic acid	2277	2279	MS,RI,O,S	500	1.26 ^b^	0.78 ^c^	1.76 ^c^	2.14 ^b^	3.04 ^b^	1.15 ^c^	3.44 ^b^	52.74 ^a^	51.18 ^a^	69.41 ^a^	Rancid, soapy.
8	Decanoic acid	2338	2356	MS,RI,O,S	130	3.25 ^c^	1.79 ^d^	4.51 ^c^	5.61 ^c^	7.69 ^c^	2.17 ^d^	8.72 ^c^	225.37 ^a^	143.67 ^b^	136.20 ^b^	Rancid, sour, fatty.
9	Undecanoic acid	2906	2910	MS,RI,O,S	1000	0 ^c^	0 ^c^	0 ^c^	0 ^c^	0.58 ^b^	1.01 ^a^	0 ^c^	0 ^c^	0.82 ^a^	1.08 ^a^	Waxy, creamy.
10	Dodecanoic acid	1450	1449	MS,RI,O,S	10,000	0.01 ^c^	0.1 ^c^	0.2 ^c^	1.02 ^b^	2.12 ^b^	2.04 ^b^	2.5 ^b^	2.01 ^b^	5.80 ^a^	3.60 ^b^	Mild, fatty, coconut.
	**Ketones**															
11	2-Pentanone	976	974	MS,RI,O,S	1380	0 ^c^	0 ^c^	0 ^c^	0 ^c^	0 ^c^	0 ^c^	0.08 ^b,c^	5.73 ^a^	0.4 ^b^	0.15 ^b^	Wine, banana, woody.
12	2-Heptanone	1181	1184	MS,RI,O,S	140	0.71 ^e^	0.62 ^e^	0.76 ^e^	0.64 ^e^	1.27 ^d^	2.37 ^d^	5.74 ^c^	115.46 ^a^	17.26 ^b^	4.37 ^c^	Fruity, coconut.
13	3-Hydroxy-2-butanone	1278	1281	MS,RI,O,S	80	2.85 ^c^	2.78 ^c^	2.72 ^c^	2.66 ^c^	3.94 ^c^	14.76 ^a^	10.58 ^ab^	6.76 ^b^	1.33 ^d^	1.13 ^d^	Buttery, creamy, milky.
14	2-Octanone	1280	1283	MS,RI,O,S	50	0 ^d^	0 ^d^	0 ^d^	0 ^d^	0 ^d^	0.22 ^c^	8.56 ^a^	1.94 ^b^	2.01 ^b^	1.84 ^b^	Woody, mushroom.
15	2-Nonanone	1385	1386	MS,RI,O,S	41	4.82 ^e^	3.94 ^e^	5.92 ^e^	6.07 ^e^	9.59 ^e^	8.73 ^e^	27.83 ^d^	610.2 ^a^	306.38 ^b^	176.91 ^c^	Fruity, buttery.
16	2-Decanone	1489	1496	MS,RI,O,S	8.3	0 ^e^	0 ^e^	0 ^e^	0 ^e^	0 ^e^	0 ^e^	1.20 ^d^	19.09 ^s^	10.82 ^b,c^	7.16 ^c^	Orange, floral, fatty.
17	2-Undecanone	1608	1615	MS,RI,O,S	5.5	18.75 ^d^	12.40 ^d^	25.91 ^d^	30.65 ^c^	38.72 ^c^	17.92 ^d^	54.15 ^c^	785.93 ^a^	472.95 ^b^	398.46 ^b^	Fruity, creamy floral.
	**Alcohols**															
18	2-Nonanol	1521	1528	MS,RI,O,S	58	1.65 ^b^	1.60 ^b^	1.46 ^b^	1.85 ^b^	1.68 ^b^	1.42 ^b^	1.53 ^b^	3.81 ^a^	1.77 ^b^	1.48 ^b^	Creamy, citrus, orange.
19	Benzene ethanol	1915	1922	MS,RI,O,S	560	0.02 ^d^	0.03 ^d^	0.06 ^d^	0.06 ^d^	0.12 ^d^	0.02 ^d^	0.85 ^c^	3.45 ^a^	1.14 ^b,c^	0.61 ^c^	Floral, rose, bready.
	**Esters**															
20	Hexanoic acid, ethyl ester	1032	1041	MS,RI,O,S	50	0 ^d^	0 ^d^	0 ^d^	0 ^d^	0 ^d^	0.14 ^d^	200.71 ^c^	544.61 ^b^	645.39 ^a^	224.83 ^c^	Pineapple, banana.
21	Heptanoic acid, ethyl ester	1133	1131	MS,RI,O,S	300	0 ^d^	0 ^d^	0 ^d^	0 ^d^	0 ^d^	0 ^d^	0 ^d^	3.23 ^b^	8.11 ^a^	1.56 ^c^	Fruity, pineapple.
22	Hexanoic acid, butyl ester	1157	1161	MS,RI,O,S	1000	0 ^d^	0 ^d^	0 ^d^	0 ^d^	0 ^d^	0.06 ^c^	0.07 ^c^	4.41 ^ab^	5.17 ^a^	2.67 ^b^	Winey, berry, soapy.
23	Octanoic acid, ethyl ester	1216	1245	MS,RI,O,S	960	0 ^d^	0 ^d^	0 ^d^	0 ^d^	0 ^d^	0 ^d^	3.11 ^c^	36.05 ^ab^	43.72 ^a^	23.66 ^b^	Wine, brandy, pear.
24	Nonanoic acid, ethyl ester	1231	1246	MS,RI,O,S	390	0 ^c^	0 ^c^	0 ^c^	0 ^c^	0 ^c^	0.13 ^c^	4.41 ^a^	1.89 ^b^	2.31 ^b^	0 ^c^	Fruity, rose, waxy.
25	Hexanoic acid hexyl ester	1262	1256	MS,RI,O,S	0.5	0 ^d^	0 ^d^	0 ^d^	0 ^d^	0 ^d^	0 ^d^	30.44 ^c^	76.34 ^b^	221.40 ^a^	83.39 ^b^	Herbal, vegetable.
26	Decanoic acid, ethyl ester	1338	1341	MS,RI,O,S	530	0 ^d^	0.01 ^d^	0.01 ^d^	0.02 ^d^	0.04 ^d^	0 ^d^	3.87 ^c^	56.29 ^b^	81.22 ^a^	50.06 ^b^	Oily, brandy, apple.
27	Undecanoic acid, ethyl ester	1349	1451	MS,RI,O,S	3	0 ^d^	0 ^d^	0 ^d^	0 ^d^	0 ^d^	5.76 ^c^	134.45 ^a^	52.00 ^b^	59.02 ^b^	0 ^d^	Soapy, waxy, fatty.
28	*δ*-hexanolactone	1419	1427	MS,RI,O,S	20	0 ^b^	0 ^b^	0 ^b^	0 ^b^	0 ^b^	0 ^b^	0 ^b^	4.12 ^a^	0 ^b^	0 ^b^	Herbal, coconut.
29	Dodecanoic acid ethyl ester	1421	1422	MS,RI,O,S	2	0 ^a^	0 ^a^	0 ^a^	0 ^a^	0 ^a^	0 ^a^	110.49 ^a^	119.95 ^a^	225.84 ^a^	123.15 ^a^	Creamy, coconut.
30	*γ*-octalactone	1433	1440	MS,RI,O,S	20	0 ^b^	0 ^b^	0 ^b^	0.24 ^b^	0.47 ^b^	0 ^b^	0 ^b^	10.94 ^a^	12.07 ^a^	10.01 ^a^	Fatty, buttery, milky.
31	*δ*-octalactone	1532	1526	MS,RI,O,S	0.42 ^a^	0 ^f^	0 ^f^	0 ^f^	0 ^f^	0 ^f^	23.14 ^d^	11.49 ^e^	142.11 ^b^	177.55 ^a^	113.55 ^c^	Buttery, fruity.
	**Others**															
32	Styrene	1513	1528	MS,RI,O,S	65 ^a^	6.83 ^e^	7.16 ^e^	7.00 ^e^	7.31 ^e^	7.16 ^e^	10.66 ^d^	4.87 ^f^	53.63 ^a^	36.96 ^b^	21.94 ^c^	Balsam, floral, plastic.
33	Benzaldehyde	1247	1254	MS,RI,O,S	750 ^c^	0.43 ^c^	0.46 ^c^	0.47 ^c^	0.51 ^c^	0.53 ^c^	0.57 ^c^	0.44 ^c^	2.16 ^a^	0.89 ^bc^	0.59 ^c^	Bitter, almond, cherry.

1. Calc = calculated. 2. Ref = reference. 3. Method of identification: MS = mass spectrum comparison using NIST 14. library; RI = retention index. O = odor description; S = standard compound.

## Data Availability

Data are contained within the article.

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
