# Peer review of "Changes in Key Aroma Compounds and Esterase Activity of Monascus-Fermented Cheese across a 30-Day Ripening Period"

_foods, 2022, doi:10.3390/foods11244026_

Round 1
Reviewer 1 Report
Excellent study that comprehensively evaluates the effect of enzymatic activity on cheese characteristics.
However, some changes to the text are necessary to make the paper clearer and more complete.
1. Improve summary.
2. Page 77: cite the location of the cheese-producing farm.
3. The method for determining volatiles was developed by the authors? If not, refer.
4. Reference the methodology of esterase activity.
5. Cite other works for comparison during the text.
6. Start by discussing the results of the analysis, and then present the figures, in the studies of enzymatic activity and analysis of sensory properties.
Author Response
Dear reviewer:
We are very grateful for your comments on the manuscript. According to your advice, we tried our best to amend the relevant part. All of your questions were answered below. And here we list the changes and marked them in red in the revised paper.
Should you have any questions, please contact us without hesitation.
Once again, thank you very much for your comments and suggestions.
Yours Sincerely,
Bei
- Improve summary.
Response: Thank you for your valuable guidance, the conclusion section has been reworked. The previous simple listing of the results has been corrected, and now the results obtained from the experiment are summarized.
- Page 77: cite the location of the cheese-producing farm.
Response: Farm locations are now clearly marked (line 78).
- The method for determining volatiles was developed by the authors? If not, refer.
Response: The method for determining volatile flavor compounds is obtained from comprehensive references and previous research of the research group, and the references have been cited now (line 100).
- Reference the methodology of esterase activity.
Response: Thank you for your valuable guidance, and references are now cited (line 140).
- Cite other works for comparison during the text.
Response: The results of this study on Monascus-fermented cheese have been compared with those of other researchers on Monascus-fermented cheese. (line 202, 268)
- Start by discussing the results of the analysis, and then present the figures, in the studies of enzymatic activity and analysis of sensory properties.
Response: Thank you for helping me straighten out my writing. The order of discussion and data has been adjusted (Sections 3.2 and 3.3).

Reviewer 2 Report
I suggest the remotion of “Monascus-fermented cheese (MC)” because this keyword is in the title.
In my opinion the best keyword to “quantitative descriptive analysis” is sensory properties or sensory analysis. You will have more chances in data science libraries.
I suggest standardization about the named, in my opinion you can use “mold-ripened cheeses”.
Could you explain more in your methodology about this technique “microextraction-Arrow”
Is this value r low? (|r| > 0.6, P < 0.05). Maybe, you can available just compounds with >0.8. In your discussion you signalize that. Your heat maps will be cleaner.
What are differences between ripened (title and lines 49/56), aging, and matured (lines 20/ 21/64/68/69/73)? If it is the same, please standardized with the same name.
Cheese manufacture – are there references before this study? Is it based in some references?
Figure 1. RAW cheese in all text (include in the title or remove to schema). The milk was pasteurized (text) or sterilization (schema)? Thus, the milk is not raw to produce cheese. Ok?
The process of making (name all process described in this figure) Monascus-fermented cheeses and control cheeses.
Line 99 -If you would like still with “silicone septum” information, please added all information about these septum’s.
Line 101. Verified this information “(DVB/CWR/PDMS)” The correct to me is DVB/CAR/PDMS.
Line 100. “equilibrated for 30 min at 60°C” – It is not so high temperature?
Line 102 and 103 “was exposed to the sample headspace for 30 min under the original experimental conditions (the extraction conditions were optimized by pre-experiments)”. Is original experimental conditions are another reference? or the equilibrated parameters (30 min at 60°C)?. I would like to see the results about optimized pre-experiments, maybe you could insert these dates in supplementary material or use the reference if these dates are published before.
Line 104. Arrow fiber – Is it similar to Holder?
Line 108 Extraction fiber is similar to Arrow fiber or Holder? If yes, please use the same name.
Line 111. Why you use 1.2 mL helium for minute? Is it not so fast?
Line 117 to 129 . Congratulations nice methodology.
2.4 Sensory evaluation
Did you have some photos about your test and your references? If yes, please, include.
I not found in material and methods: information’s about how did you identified volatile compounds? Did you use some alkanes to calculate LRI? or Did you use some library references? or Authentic compounds? Did you submit this assay for an ethical committee? (RIB)
I would like to see a Tukey test to Table 1, figure 1 and figure 4 (c and d).
I liked your discussion, but some points could be improved.
The best point to maturation and sensory properties (QDA and GCO-MS) it is after 20 days of ripeness. The main odor-active in this study was (2,3,4,5,6,7, 21,22,26,27,29 – numbers to table 1 compounds). Please lock the low threshold value and the high concentration to these compounds in MC-20 and MC-30. Some of than have off-flavors (rancid, pungent, fatty…) maybe these could be the reason to low values for cheese characterization showed in figure 4d (20-30days for MC). It is so important you show your standards to QDA, maybe fruit and wine could be mischaracterized cheese.
Conclusion is not appropriate. Conclusion is a very short topic. Just answered your objective. Maybe you could include some information in the discussion last paragraph or include a new topic to “This work enriches the knowledge of the evolution of volatile aroma compounds….”.
Author Response
Dear reviewer:
We are very grateful for your comments on the manuscript. According to your advice, we tried our best to amend the relevant part. All of your questions were answered below. Please see the attachment.
Should you have any questions, please contact us without hesitation.
Once again, thank you very much for your comments and suggestions.
Yours Sincerely,
Bei

Round 2
Reviewer 2 Report
Line 102 and 103 I would like to see the results about optimized pre-experiments, maybe you could insert these dates in supplementary material or use the reference if these dates are published before. “was exposed to the sample headspace for 30 min under the original experimental conditions (the extraction conditions were optimized by pre-experiments)”.
Table 1. Include two columns (LRI experimental and LRI literature) and include an information present just in supplementary table (identification-last column).
2.4 Sensory evaluation
Please, include some photos or composition about your sensory test and your references (standards used in QDA).
What statistical test did you use? Different letters in the same row correspond to statistically significant differences (P < 0.05). Tukey? Include in your statistic topic.
Conclusion still not appropriate. Conclusion is a very short topic. Just answered your objective. Maybe you could include some information in the last paragraph of discussion or include a new topic to “This work enriches the knowledge of the evolution of volatile aroma compounds….”.
Author Response
Dear reviewer:
Thank you again for your comments and suggestions on the manuscript. According to your advice, we tried our best to amend the relevant part. All of your questions were answered below. Please see the attachment.
Should you have any questions, please contact us without hesitation.
Yours Sincerely,
Bei
